# Perceived Parental Attitudes Are Indirectly Associated with Consumption of Junk Foods and Sugar-Sweetened Beverages among Chinese Adolescents through Home Food Environment and Autonomous Motivation: A Path Analysis

**DOI:** 10.3390/nu13103403

**Published:** 2021-09-27

**Authors:** Nan Qiu, Justin B. Moore, Yechuang Wang, Jialin Fu, Kai Ding, Rui Li

**Affiliations:** 1Department of Healthcare Management, School of Health Sciences, Wuhan University, Wuhan 430071, China; 2013302170051@whu.edu.cn (N.Q.); ywang20@whu.edu.cn (Y.W.); fjl0708@whu.edu.cn (J.F.); 2021203050024@whu.edu.cn (K.D.); 2Department of Implementation Science, Division of Public Health Sciences, Wake Forest School of Medicine, Winston-Salem, NC 27101, USA; jusmoore@wakehealth.edu

**Keywords:** perceived parental attitudes, junk foods, sugar-sweetened beverages, home food environment, autonomous motivation

## Abstract

This study aimed to use path analysis to determine the association between perceived parental attitudes toward restricting junk food (JF)/sugar-sweetened beverage (SSB) intake and JF/SSB consumption among Chinese adolescents, and whether JF/SSB availability in the home environment and autonomous motivation of adolescents mediated the association. A cross-sectional survey was conducted using questionnaires adapted from the Family Life, Activity, Sun, Health, and Eating (FLASHE) Study to collect data on 3819 participants with an average age of 14.7 years (SD = 1.7). Spearman correlations and path analysis were performed. It was found that perceived parental attitudes were not directly associated with adolescents’ JF/SSB consumption frequency, but indirectly related to them through JF/SSB availability in the home environment and autonomous motivation of adolescents. When parents held a less positive attitude toward JF/SSB consumption and kept less JFs/SSBs at home, youth displayed more autonomous motivation for restricting JF/SSB intake and consumed fewer JFs/SSBs.

## 1. Introduction

Junk foods (JFs) are defined as high energy foods with minimal nutritional value, higher content of saturated fat, and more salt and/or sugar. Generally these include fried potato products, potato crisps, snacks, sweet and salty biscuits/cakes/doughnuts, confectionary, and ice cream [1]. Sugar-sweetened beverages (SSBs) refer to drinks with added sugar such as soda, sweetened fruit drinks, and energy drinks [2]. Globally, JFs and SSBs are popular among all age groups [3,4]. However, JFs and SSBs are considered unhealthy foods among health professionals. It has been shown that JF and SSB consumption is associated with a high prevalence of obesity [5], and contribute to hypertension [6], type 2 diabetes, and cardiovascular disease [7].

A low-nutrient/high-energy diet in adolescents has become one of the most important nutrition-related concerns globally [8,9]. Adolescence is a significant period of physical, social, and emotional development where dietary patterns are being established [10]. JFs and SSBs are inexpensive and easily available making them particularly attractive to young people [11]. A study highlighted the negative impact of JFs and SSBs on adolescents’ brain function, showing that JFs and SSBs can lead to cognitive impairments and changes in reward processing [11]. Additionally, high consumption of JFs and SSBs along with low consumption of fruits and vegetables is associated with an increased risk of depression among adolescents [12]. Moreover, it has been shown that high consumption of JFs and SSBs significantly contributes to the high prevalence of overweight and obesity in children and adolescents [13]. Identifying the factors related to limiting consumption of JFs and SSBs as an effort to reduce the growing prevalence of obesity has attracted a growing public health concern.

One’s environment can influence personal food intake [14]. Parents play a critical role for the physical and psychosocial environment of their children [15,16]. Parents can affect their children’s eating patterns through their attitudes, behaviors, and feeding styles [17]. It has been shown that parental feeding attitudes are significantly related to food intake and eating motivation of children [18]. Parents promote healthy eating by providing advice on food selection and home food environment [19]. However, these studies mainly focused on the influence of parental attitudes on promoting healthy food intake, such as fruits and vegetables, rather than on restricting JF and SSB consumption [20,21]. It has been found that food availability in the home environment is related to food consumption among 4th grade children [22]. In fact, the more frequently JFs and SSBs are provided in the home environment, the more likely younger children are to consume them [13]. However, the majority of the prior research has been conducted in North American and European families, but the extent these findings apply to Chinese families is unclear.

According to self-determination theory, motivation, including autonomous motivation and controlled motivation, are predictors of human behaviors [23]. Autonomous motivation is the concept that people have recognized the value of an activity and have integrate it into their self-awareness [24]. It has been shown that when people are autonomously motivated, their healthy behavior changes are more effective and lasting [25]. Adolescents’ autonomous motivation on SSB consumption could significantly influence their beverage intake [26]. External factors can also affect autonomous motivation and personal behaviors [23]. For example, children who had autonomy-supportive parents were more intrinsically motivated than their peers with controlling parents [27]. This could be explained by that parents’ encouragement and modeling would increase autonomous motivation of adolescents rather than mandate and supervision [28].

Therefore, this study was designed to employ path analysis to elucidate the association between perceived parental attitudes toward restricting JF/SSB intake and JF/SSB consumption among Chinese adolescents, and whether JF/SSB availability in the home environment and autonomous motivation of adolescents mediated the association. We supposed that parental attitudes were associated with JF/SSB consumption among adolescents through JF/SSB availability in the home environment and autonomous motivation of adolescents. We hypothesized their associations as presented in Figure 1.

## 2. Methods

### 2.1. Participants

A cross-sectional study was conducted at a high school in Wuhan, China in October 2019. Data collection procedures were carried out in accordance with the Declaration of Helsinki and approved by the Wuhan University Ethics Board (ethical approval code: 2019YF2056) and the local school district administrators. Before participation, informed consent forms were distributed to all adolescents in this school (*n* = 4519) aged 10–20 years to obtain parental consent [29]. A total of 4027 adolescents were eligible and consented to participate in the study. Participants filled in the questionnaires by themselves and all of them returned the questionnaires. Questionnaires that had missing data in JF/SSB consumption frequency were excluded (*n* = 208). A total of 3819 participants were included in the analyses.

### 2.2. Measure

Participants reported their age, gender, grade, ethnicity, height and weight, parents’ education level, and household monthly income by themselves. Body mass index (BMI) was calculated as weight (kg) divided by height (m^2^). Body weight status was classified based on Centers for Disease Control and Prevention’s sex-specific 2000 BMI-for-age growth charts as underweight (BMI < 5th percentile, BMI z-score ≤ −2), healthy weight (5th percentile ≤ BMI < 85th percentile, −2 < BMI z-score ≤ 1), overweight (85th percentile ≤ BMI < 95th percentile, 1 < BMI z-score < 2), and obesity (BMI ≥ 95th percentile, BMI z-score ≥ 2) [28].

Questionnaires from the Family Life, Activity, Sun, Health, and Eating (FLASHE) Study were used to collect information on the consumption frequency of JFs/SSBs, perceived parental attitudes, JF/SSB availability in the home environment, and autonomous motivation. The FLASHE Study was developed by The National Cancer Institute with cognitive testing and usability testing [30]. Source information and full survey wording can be found on the FLAHSE website [31]. We translated the questionnaire into Chinese and conducted the reliability and validity test: Cronbach’s Alpha_JF intake_ = 0.86; Cronbach’s Alpha_SSB intake_ = 0.80; Cronbach’s Alpha_Perceived parental attitudes_ = 0.70; Cronbach’s Alpha_JF availability in the home environment_ = 0.68; Cronbach’s Alpha_Autonomous motivation_ = 0.69; Kaiser–Meyer–Olkin = 0.86, P Bartlett < 0.001. Response options of JF/SSB intake, based on the consumption frequency in past 7 days, were converted to a daily frequency (e.g., Never = 0; 1–3 times in past 7 days = 0.3; 4–6 times in past 7 days = 0.7) [28]. Detail items and responses of perceived parental attitudes, JF/SSB availability in the home environment and adolescents’ autonomous motivation could be found in Appendix A.

### 2.3. Analyses

All statistical analysis were performed using SPSS 21.0 (IBM, Armonk, NY, USA). Confidence intervals of indirect and direct effects were calculated using the PROCESS (Model 6) for SPSS developed by Hayes (2017). As the missing responses met the assumption for missing at random, a multiple imputation method was used to fill in the null value of the analytical sample including 3819 participants. Descriptive statistics were used to describe participant characteristics, JF/SSB consumption frequency, perceived parental attitudes, JF/SSB availability in the home environment and adolescents’ autonomous motivation. Categorical variables were represented by frequency and percentage, while continuous variables were represented by mean and standard deviation (normal) or median and inter-quartile range (non-normal). Spearman correlation analysis was used to test the relationship among the variables. If perceived parental attitudes, JF/SSB availability in the home environment, autonomous motivation of adolescents and JF/SSB consumption frequency among adolescents were all significantly correlated with each other, the path analysis would be conducted to verify the hypothesis. The path analysis was adjusted for adolescents’ age, gender, ethnicity, BMI z-score, parents’ educational level and household monthly income. A bootstrapping procedure (*n* = 5000) was performed to calculate the path coefficient and 95% confidence intervals. Statistical significance was set at two-tailed *p* < 0.05, 95% confidence intervals did not include 0.

## 3. Results

Table 1 showed the non-imputed data of participants’ characteristics. There were 3819 adolescents with an average age of 14.7 years (SD = 1.7), including 2016 (52.7%) males and 1758 (40.0%) junior high school students. 798 (20.8%) students were overweight or obese. Most (98%) of the students’ ethnicity were Han. Most participants had a household monthly income between 5000 and 20,000. More than half of students consumed JFs at least once a day, and less than a half of adolescents reported that they consumed SSBs more than once a day.

The descriptive statistics of JF/SSB consumption frequency, scores of perceived parental attitudes toward limiting JF/SSB intake, JF/SSB availability in the home environment and autonomous motivation of adolescents to restrict JFs/SSBs could be viewed in Table 2. None of these variables were normally distributed, so they were represented by the median and interquartile range.

The results of the correlation analysis to examine the relationship between JF/SSB consumption, perceived parental attitudes toward limiting JF/SSB intake, JF/SSB availability in the home environment and autonomous motivation of adolescents were shown in Table 3 and Table 4. There were significant correlations among JF/SSB consumption among adolescents, perceived parental attitudes, JF/SSB availability in the home environment, and autonomous motivation of adolescents (*p* < 0.01).

The models of path analysis were presented in Figure 2 and Figure 3. Perceived parental attitudes were negatively associated with the JF/SSB availability in the home environment (b_JFs_ = −0.203, 95% CI_JFs_ = −0.241, −0.166; b_SSBs_ = −0.215, 95% CI_SSBs_ = −0.261, −0.169), but positively related to adolescents’ autonomous motivation (b_JFs_ = 0.266, 95% CI_JFs_ = 0.222, 0.310; b_SSBs_ = 0.273, 95% CI_SSBs_ = 0.230, 0.315). JF/SSB availability in the home environment was negatively correlated with adolescents’ autonomous motivation (b_JFs_ = −0.054, 95% CI_JFs_ = −0.102, −0.011; b_SSBs_ = −0.048, 95% CI_SSBs_ = −0.084, −0.012), while positively correlated with JFs/SSBs intake (b_JFs_ = 1.056, 95% CI_JFs_ = 0.869, 1.240; b_SSBs_ = 0.605, 95% CI_SSBs_ = 0.551, 0.659). Autonomous motivation of adolescents was negatively associated with JF/SSB consumption (b_JFs_ = −0.096, 95% CI_JFs_ = −0.176, −0.017; b_SSBs_ = −0.152, 95% CI_SSBs_ = −0.209, −0.095).

Table 5 showed that the direct effect of perceived parental attitudes on JF intake among adolescents was not significant, but the indirect effect through JF availability in the home environment (b = −0.214, 95% CI = −0.274, −0.160) and autonomous motivation (b = −0.026, 95% CI = −0.054, −0.012) was significant and the total effect was b = −0.242 (95% CI = −0.348, −0.135). Results were similar for SSB intake. The indirect effect of perceived parental attitudes on SSBs consumption through SSBs availability in home environment (b = −0.130, 95% CI = −0.167, −0.097) and autonomous motivation (b = −0.042, 95% CI = −0.063, −0.021) was significant and the total effect was b = −0.221 (95% CI = −0.298, −0.145). Hence, JF/SSB availability in the home environment and autonomous motivation of adolescents showed a complete mediation effect in the relationship between perceived parental attitudes and consumption of JFs/SSBs among adolescents.

## 4. Discussion

The current study examined the relationship between JF/SSB intake and perceived parental attitudes, JF/SSB availability in the home environment and autonomous motivation of adolescents, and identified the pathway that how perceived parental attitudes could potentially influence adolescents’ JF/SSB consumption.

A notable finding was that perceived parental attitudes was not directly associated with the JF/SSB consumption among adolescents, but indirectly related to it through availability of JFs/SSBs in home environment and autonomous motivation of adolescents. Perceived parental attitudes toward restricting their children’s JF/SSB consumption were negatively associated with JF/SSB availability in the home environment, which meant that when parents had a willing on limiting adolescents’ JF/SSB consumption, parents would protect their children from exposure to JFs/SSBs. It was found that adolescents would not keep SSBs at home if they perceived disagreement from parents on SSB intake [32]. Furthermore, we found that availability of JFs in the home environment was directly and positively correlated with consumption of JFs among adolescents, which was in agreement with a previous study that suggested adolescents with higher frequent availability of JFs at home reported higher consumption of JFs [33]. This relationship was also found when studying the association between SSB intake and SSB availability in the home environment. It was reported that most SSBs were consumed in the home environment among children and adolescents [34,35]. Hence the availability of SSBs in the home environment is an important factor and positively correlated with SSB intake among adolescents [36]. In addition, availability of JFs/SSBs in home environment was directly and negatively associated with autonomous motivation of adolescents for avoiding JF/SSB consumption, which supported a previous study demonstrating that adolescents were more inclined to eat food that was available and easily accessible, and they tended to eat greater quantities when larger portions were provided [37]. These findings suggested that a healthy home environment with less available unhealthy food was critical for adolescents [38].

Perceived parental attitudes toward restricting consumption of JFs/SSBs were positively related to autonomous motivation of adolescents on limiting JF/SSB intake, and when adolescents were more motivated to stay away from JFs and SSBs, they reported consuming fewer JFs and SSBs. This pattern has also been found in American teenagers [39]. There is no doubt that parents play an essential role in supporting psychological needs of children [40]. What is more, non-compulsory support can strengthen adolescents’ autonomous motivation, while regulation may weaken it [39,41]. A previous study on the relationship between parental support and autonomous motivation indicated that perceived parental attitudes and behaviors regarding diet could affect autonomous motivation of adolescents to choose healthy food and avoid JFs resulting in more successful weight loss attempts [42].

Overall, adolescents who perceived less positive parental attitudes towards JFs/SSBs, consumed JFs and SSBs less frequently. This finding was consistent with the FLASHE study in the American adolescents [43]. Therefore, interventions that make parents aware of the unhealthy effects of JF/SSB excessive consumption are needed so as to ensure the informed decisions about storing JFs/SSBs at home can be made. Moreover, some studies suggested that the mixed feeding practices used by parents, such as limiting the availability of JFs/SSBs and being a role model to avoid consuming JFs/SSBs, to reduce their children’s JF/SSB intake might be more effective than mandatory control practices, such as deciding how much JFs/SSBs to eat [18,28]. Additionally, the evidence provided in current study supports the use of self-determination theory as a framework to study the interaction between parental practices and adolescent dietary behaviors [23].

Although prior studies found that parental attitudes were related to consumption of JFs/SSBs among adolescents, few study has assessed the pathway from perceived parental attitudes to adolescent JF/SSB intake as mediated by home food availability and the youth’s autonomous motivation. However, there are limitations to be acknowledged when interpreting this study. Firstly, though we had a relatively large sample size, participants in this study were recruited from one school, which limited the representativeness of the sample. Secondly, as a cross-sectional study, findings of this research could only support associations but not causality.

## 5. Conclusions

The findings from this study shed some light on the associations between perceived parental attitudes and JF/SSB consumption among Chinese adolescents. Perceived parental attitudes were indirectly associated with JF/SSB consumption through JF/SSB availability in the home environment and autonomous motivation of adolescents. When parents have less positive attitudes toward JFs/SSBs and keep less JFs/SSBs at home, youth report more autonomous motivation for restricting JFs/SSBs and consume fewer JFs/SSBs.

## Figures and Tables

**Figure 1 nutrients-13-03403-f001:**
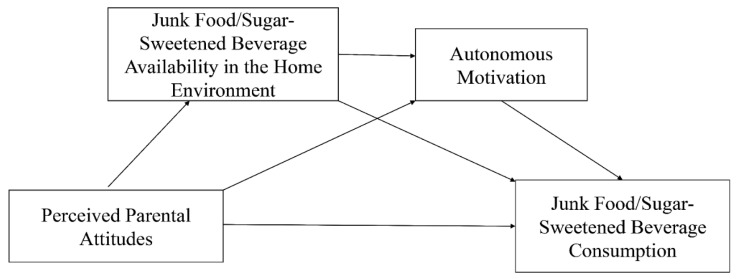
A path model to explain junk food and sugar-sweetened beverage consumption.

**Figure 2 nutrients-13-03403-f002:**
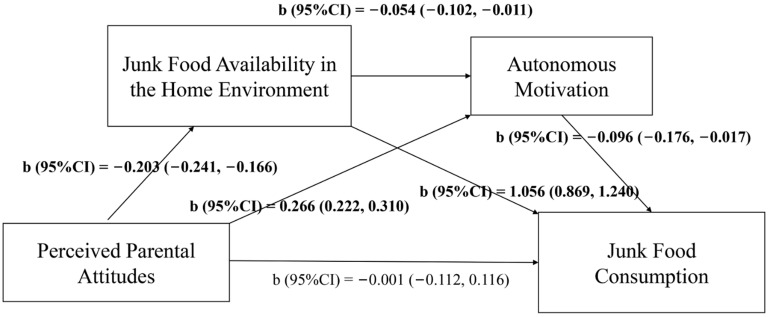
Path model of junk food consumption. Note: b, coefficient; CI, confidence interval; Highlighted format indicated that b was statistically significant.

**Figure 3 nutrients-13-03403-f003:**
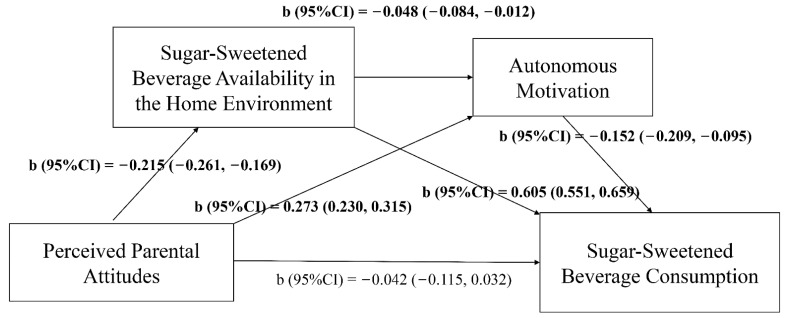
Path model of sugar-sweetened beverage consumption. Note: b, coefficient; CI, confidence interval; Highlighted format indicated that b was statistically significant.

**Table 1 nutrients-13-03403-t001:** Characteristics of Participants *.

	*n*	%
Age (mean, SD)	14.7	1.7
*Gender*		
Male	2016	52.7
Female	1762	46.1
*Educational level*		
Junior high school	1758	40.0
Senior high school	2061	60.0
*Weight status*		
Underweight	95	2.4
Healthy weight	2926	76.6
Overweight or obese	798	20.8
*Ethnicity*		
Han	3745	98.0
Ethnic minority	35	1.0
*Household monthly income*		
Below 5000 RMB	515	13.4
5000 RMB–10,000 RMB	1703	44.5
10,000 RMB–20,000 RMB	915	23.9
More than 20,000 RMB	376	9.7
*Food consumption frequency*		
Junk food intake = 0 time/day	265	6.9
Junk food intake < 1 time/day	1273	33.3
Junk food intake ≥ 1 time/day	2281	59.8
Sugar-sweetened beverage intake = 0 time/day	411	10.8
Sugar-sweetened beverage intake < 1 time/day	1783	46.7
Sugar-sweetened beverage intake ≥ 1 time/day	1625	42.5

* Non-imputed data, so the sample size varies by variable due to missing data; SD, standard deviation.

**Table 2 nutrients-13-03403-t002:** Descriptive statistics of junk food/sugar-sweetened beverage consumption frequency, perceived parental attitudes, home food environment, and autonomous motivation.

	Minimum	Maximum	Median (Interquartile Range)
Junk food consumption frequency (times/day)	0	18	1.2 (0.6, 2)
Sugar-sweetened beverage consumption frequency (times/day)	0	12	0.6 (0.3, 1.4)
Perceived parental attitudes	1	5	3.25 (3, 3.75)
Junk food availability in the home environment	1	5	2 (1.5, 2.5)
Sugar-sweetened beverage availability in the home environment	1	5	2 (2, 3)
Autonomous motivation	1	5	4 (3, 4.5)

**Table 3 nutrients-13-03403-t003:** The correlations among junk food consumption, perceived parental attitudes, junk food availability in the home environment and autonomous motivation.

	Junk Food Consumption	Perceived Parental Attitudes	Junk Food Availability in the Home Environment	Autonomous Motivation
Junk food consumption	1	−0.136 *	0.412 *	−0.095 *
Perceived parental attitudes		1	−0.199 *	0.212 *
Junk food availability in the home environment			1	−0.115 *
Autonomous motivation				1

* *p* < 0.01.

**Table 4 nutrients-13-03403-t004:** The correlations among sugar-sweetened beverage consumption, perceived parental attitudes, sugar-sweetened beverage availability in the home environment and autonomous motivation.

	Sugar-Sweetened Beverage Consumption	Perceived Parental Attitudes	Sugar-Sweetened Beverage Availability in the Home Environment	Autonomous Motivation
Sugar-sweetened beverage consumption	1	−0.144 *	0.391 *	−0.160 *
Perceived parental attitudes		1	−0.156 *	0.212 *
Sugar-sweetened beverage availability in the home environment			1	−0.092 *
Autonomous motivation				1

* *p* < 0.01.

**Table 5 nutrients-13-03403-t005:** The result of the path analysis.

	b	95% CI	
*Effect of perceived parental attitudes on junk foods*			
Total effect	−0.242	−0.348	−0.135
Direct effect	−0.001	−0.104	0.102
Total indirect effects	−0.241	−0.306	−0.182
Perceived parental attitudes—junk food availability in the home environment—junk foods	−0.214	−0.274	−0.160
Perceived parental attitudes—autonomous motivation—junk foods	−0.026	−0.054	−0.012
Perceived parental attitudes—junk food availability in the home environment—autonomous motivation—junk foods	−0.001	−0.002	−0.0003
*Effect of perceived parental attitudes on sugar-sweetened beverages*			
Total effect	−0.221	−0.298	−0.145
Direct effect	−0.042	−0.115	0.032
Total indirect effects	−0.180	−0.221	−0.141
Perceived parental attitudes—sugar-sweetened beverage availability in the home environment—sugar-sweetened beverages	−0.130	−0.167	−0.097
Perceived parental attitudes—autonomous motivation—sugar-sweetened beverages	−0.042	−0.063	−0.021
Perceived parental attitudes—sugar-sweetened beverage availability in the home environment—autonomous motivation—sugar-sweetened beverages	−0.008	−0.015	−0.001

b, coefficient; 95% CI, 95% confidence interval.

## Data Availability

The data presented in this study are available on request from the corresponding author. The data are not publicly available due to privacy restrictions.

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
