# Peer review of "Perceived Parental Attitudes Are Indirectly Associated with Consumption of Junk Foods and Sugar-Sweetened Beverages among Chinese Adolescents through Home Food Environment and Autonomous Motivation: A Path Analysis"

_nutrients, 2021, doi:10.3390/nu13103403_

Round 1

Reviewer 1 Report

This is a really interesting topic your research is on! However, there are some issues before publication, which are presented below.

Line 56: replace “this studies” with these studies”

Line 90: It is stated that adolescents were considered persons aged 10 – 20 years old. A reference is needed here, as adolescence sometimes is considered to be from age 12 to 18, 10 to 19, 10 to 21, etc.

Table 1: I would be helpful for the reader to state whether the frequency of junk food or SSB you depict is 1/ day, week etc. (1st column).

Table 2: Please add values in brackets in column 1 for all variables. For instance, Junk food consumption (times/ day). Also, for those variables that are categorical (such as parental behaviors) you need to present descriptive statistics as percentages of answers for each behavior.

Regarding methods, on what tool or rationale is your autonomous motivation assessment based?

Some phrasal concerns exist, like lines 208 – 209 “which meant that 208 when parents had a willing on limiting”. Please re-read the paper. Also some syntax errors are present, see lines 211 – 215.

Lines 225 – 228. This would be preferably moved to end of discussion as to practical implications f the findings. However, as this is a cross-sectional study, be very careful with the interpretation of your findings, as causality may not be revealed as you also acknowledge on lines 157 – 258.. So, you may probably need to omit that.

Line 240. Rephrase this line as it does not depict what you describe in the rest sentence.

Regarding results:

1) where there any differences in findings between girls and boys? Please depict that.

2) As this is a large age group, did you try to split the group in two or three, and run the analysis again, as there are stages in adolescence?

3) Were some parental behaviors more or less strongly associated to junk food and SSB intake?

Author Response

Dear Reviewer,

The authors thank you for your time and effort to provide us with those valuable comments. We had read them with great interest and responded them line by line as indicated below. The original comments were marked by bold font and our responses were marked by italic font.

  1. Line 56: replace “this studies” with these studies”

Corrected, we have revised the word. Please see the change highlighted in the revised manuscript.

  1. Line 90: It is stated that adolescents were considered persons aged 10 – 20 years old. A reference is needed here, as adolescence sometimes is considered to be from age 12 to 18, 10 to 19, 10 to 21, etc.

We have added the reference. Please see the change highlighted in the manuscript.

  1. Table 1: I would be helpful for the reader to state whether the frequency of junk food or SSB you depict is 1/ day, week etc. (1st column). And Table 2: Please add values in brackets in column 1 for all variables. For instance, Junk food consumption (times/day). Also, for those variables that are categorical (such as parental behaviors) you need to present descriptive statistics as percentages of answers for each behavior.

Corrected, we have added values in brackets in column 1. Please see the change highlighted in the revised manuscript. Besides, perceived parental attitudes are grade data that cannot be expressed as a percentage.

  1. Regarding methods, on what tool or rationale is your autonomous motivation assessment based?

Questionnaires from the Family Life, Activity, Sun, Health, and Eating (FLASHE) Study were used to collect information on the autonomous motivation.

  1. Some phrasal concerns exist, like lines 208 – 209 “which meant that 208 when parents had a willing on limiting”. Please re-read the paper. Also some syntax errors are present, see lines 211 – 215.

Corrected, we have revised the words. Please see the change highlighted in the revised manuscript.

  1. Lines 225 – 228. This would be preferably moved to end of discussion as to practical implications the findings. However, as this is a cross-sectional study, be very careful with the interpretation of your findings, as causality may not be revealed as you also acknowledge on lines 157 – 258. So, you may probably need to omit that.

This sentence is a suggestion for the future intervention study, and we have moved it to the end of discussion. Please see the change highlighted in the revised manuscript.

  1. Line 240. Rephrase this line as it does not depict what you describe in the rest sentence.

We have deleted some words to make this sentence more rigorous. Please see the change highlighted in the revised manuscript.

  1. Regarding results, the reviewer gave 3 interesting topics for further study:

1) Where there any differences in findings between girls and boys? Please depict that.

2) As this is a large age group, did you try to split the group in two or three, and run the analysis again, as there are stages in adolescence?

3) Were some parental behaviors more or less strongly associated to junk food and SSB intake?

In the path model, b(95%CI)gender = 0.007(-0.073, 0.059), b(95%CI)age = 0.009(-0.010, 0.028). We therefore did not perform subgroup analyses on gender and age. However, different parental behaviors would be an interesting point that we’ll focus on in our follow-up studies.

Reviewer 2 Report

Methods
Page 3, Line 129:
Please let the readers know w hat method for correlation analysis that you
have performed to test the relationship among the variables. Please mention it.
Page 3, Line 1 05 118 : The authors measured junk food consumption of Chinese adolescents using the questionnaire from FLASHE study. The junk foods listed in Table S1 are typical junk foods in USA but they might not necessarily represent the junk foods in Wuhan, China. I’m wondering whether some participants even didn't know the name of the junk food items listed in Table S1. This might underestimate junk food consumption of the participants.

Results
Table 1: In the Method, the authors wrote A total of 3819 participants were
included in the analyses ””, however, in Table 1, 2016 males and 1762 females are not reaching to 3819 participants

Discussion
Page 9 , Line 2 4 9: ……. support the use of self determination theory
Reference No. should be given for self determination theory although the
authors have mentioned it in Introduction section.

Author Response

Dear Reviewer,

The authors thank you for your time and effort to provide us with those valuable comments. We had read them with great interest and responded them line by line as indicated below. The original comments were marked by bold font and our responses were marked by italic font.

Methods

  1. 1. Page 3, Line 129:
    Please let the readers know what method for correlation analysis that you have performed to test the relationship among the variables. Please mention it.

Corrected, we have added “Spearman correlation” in the analyses, please see the change highlighted in the revised manuscript.

2. Page 3, Line 105-118: The authors measured junk food consumption of Chinese adolescents using the questionnaire from FLASHE study. The junk foods listed in Table S1 are typical junk foods in USA but they might not necessarily represent the junk foods in Wuhan, China. I’m wondering whether some participants even didn't know the name of the junk food items listed in Table S1. This might underestimate junk food consumption of the participants.

We have considered this question when translating the questionnaire, and used the junk food and sugar-sweetened beverage in Wuhan, China.

Results

1. Table 1: In the Method, the authors wrote “a total of 3819 participants were included in the analyses”, however, in Table 1, 2016 males and 1762 females are not reaching to 3819 participants

Table 1 was non-imputed data, so the sample size varies by variable due to missing data, please see the note.

Discussion
1. Page 9, Line 2 4 9: ……. support the use of self-determination theory Reference No. should be given for self-determination theory although the authors have mentioned it in Introduction section.

Corrected, we have added the reference. Please see the change highlighted in the revised manuscript.